# Efficacy and Safety of Atezolizumab and Bevacizumab in the Real-World Treatment of Advanced Hepatocellular Carcinoma: Experience from Four Tertiary Centers

**DOI:** 10.3390/cancers14071722

**Published:** 2022-03-28

**Authors:** Vera Himmelsbach, Matthias Pinter, Bernhard Scheiner, Marino Venerito, Friedrich Sinner, Carolin Zimpel, Jens U. Marquardt, Jörg Trojan, Oliver Waidmann, Fabian Finkelmeier

**Affiliations:** 1Department of Gastroenterology, Hepatology and Endocrinology, University Hospital Frankfurt, 60590 Frankfurt, Germany; vera.himmelsbach@kgu.de (V.H.); joerg.trojan@kgu.de (J.T.); oliver.waidmann@kgu.de (O.W.); 2Division of Gastroenterology and Hepatology, Department of Internal Medicine III, Medical University of Vienna, 1090 Vienna, Austria; matthias.pinter@meduniwien.ac.at (M.P.); bernhard.scheiner@meduniwien.ac.at (B.S.); 3Liver Cancer (HCC) Study Group Vienna, Medical University of Vienna, 1090 Vienna, Austria; 4Department of Gastroenterology, Hepatology and Infectious Diseases, Otto-Von Guericke University Hospital, 39120 Magdeburg, Germany; m.venerito@med.ovgu.de (M.V.); friedrich.sinner@med.ovgu.de (F.S.); 5Department of Internal Medicine I, Johannes Gutenberg University, 55131 Mainz, Germany; carolin.czauderna@uksh.de (C.Z.); jens.marquardt@uksh.de (J.U.M.); 6Department of Medicine I, University Medical Centre Schleswig-Holstein, Campus Lübeck, 23538 Lübeck, Germany; 7University Cancer Center Frankfurt, University Hospital Frankfurt, 60590 Frankfurt, Germany; 8Frankfurt Cancer Institute, Goethe University Frankfurt/Main, 60590 Frankfurt, Germany

**Keywords:** hepatocellular carcinoma, atezolizumab, bevacizumab, real world, immunotherapy

## Abstract

**Simple Summary:**

Hepatocellular carcinoma is one of the most common cancers in the world with increasing incidence. In advanced stages, according to the Barcelona Clinic Liver Cancer (BCLC) staging defined by number, size, vessel infiltration status, and patient’s performance status, the therapy of choice is systemic therapy. For several years, the tyrosine kinase inhibitor sorafenib was the only therapeutic option. Atezolizumab and bevacizumab are administered as a combination therapy promoting PD-L1 inhibition and anti-VEGF activity, which yields synergistic effects against cancer. The IMBRAVE150 trial investigated the use of this combination therapy versus that of sorafenib and showed an increase in overall patient survival to nearly 20 months. In this work, we investigated the real-world efficacy and safety of this combination in different centers.

**Abstract:**

The combination of atezolizumab and bevacizumab (A + B) is the new standard of care for the systemic first-line treatment of hepatocellular carcinoma (HCC). However, up to now there are only few data on the safety and efficacy of A + B in real life. We included patients with advanced HCC treated with A + B as first-line therapy at four cancer centers in Germany and Austria between December 2018 and August 2021. Demographics, overall survival (OS), and adverse events were assessed until 15 September 2021. We included 66 patients. Most patients had compensated cirrhosis (n = 34; 52%), while Child–Pugh class B cirrhosis was observed in 23 patients (35%), and class C cirrhosis in 5 patients (8%). The best responses included a complete response (CR) in 7 patients (11%), a partial response (PR) in 12 patients (18%), stable disease (SD) in 22 patients (33%), and progressive disease in 11 patients (17%)**.** The median progression-free (PFS) survival was 6.5 months, while the median overall survival (OS) was not reached in this cohort (6-month OS: 69%, 12-month OS: 60%, 18-month OS: 58%). Patients with viral hepatitis seemed to have a better prognosis than patients with HCC of non-viral etiology. The real-world PFS and OS were comparable to those of the pivotal IMBRAVE trial, despite including patients with worse liver function in this study. We conclude that A + B is also highly effective in a real-life setting, with manageable toxicity, especially in patients with compensated liver disease. In patients with compromised liver function (Child B and C), the treatment showed low efficacy and, therefore, it should be well considered before administration to these patients.

## 1. Introduction

Hepatocellular carcinoma (HCC) is the most frequent malignant primary liver cancer and the third leading reason of cancer-related death [1]. Immunotherapy is active and well tolerated in patients with advanced HCC. However, due to formally negative phase 3 trials, the anti-programmed cell death protein 1 (PD-1) antibodies nivolumab and pembrolizumab have not been approved in Europe [2,3]. The IMbrave150 study investigated the combined therapy with the anti-PD-L1 antibody atezolizumab and the vascular endothelial growth factor (VEGF)-targeting antibody bevacizumab compared to treatment with sorafenib in a phase 3 trial. The trial reached its coprimary end points of improving overall survival and progression-free survival and showed a favorable quality of life in the immunotherapy arm [4,5,6]. Based on these results, atezolizumab and bevacizumab have become the new standard of care for advanced HCC and represent the first immune checkpoint inhibitor-based combination therapy approved for HCC. However, in a real-life setting, many patients at need for anticancer treatment do not fulfill all inclusion criteria of the phase 3 trials, mainly due to impaired liver function [7]. Therefore, data from regular prescription and treatment are urgently needed. Here, we analyzed data from HCC patients treated with atezolizumab/bevacizumab in four referral centers.

## 2. Materials and Methods

### 2.1. Study Design and Selection of Patients

This was a retrospective study of patients treated with atezolizumab and bevacizumab s first-line therapy across four academic hospitals in Germany and Austria.

All patients with confirmed HCC treated with atezolizumab and bevacizumab in the individual centers between December 2018 and August 2021 were included into the analysis. Several patients were treated before the EMA approval, as a result of exceptional approvals by health care insurance companies to cover the costs.

Patients’ data including history of the disease, treatment course, laboratory results, radiological data, and follow-up were collected retrospectively from patients’ files.

The study was performed in accordance with the 1975 Declaration of Helsinki. The retrospective analysis was approved by the local Ethics Committee (SGI03/18, Amendment 01/19) as well as by the Ethics Committees of the individual centers.

### 2.2. Assessments

Electronic hospital charts were retrospectively analyzed for baseline demographic data and laboratory results.

Radiological response was recorded by computed tomography (CT) or magnetic resonance imaging (MRI) at baseline, 6–12 weeks after treatment initiation, and about every 2–3 months thereafter according to the local guidelines. Tumor response was assessed according to the Response Evaluation Criteria in Solid Tumors (RECIST) V1.1 [8] or modified RECIST [9] (according to centers’ preference). Side effects were recorded at every visit and graded according to the Common Terminology Criteria for Adverse Events (CTCAE) version 4.0 [10] or 5.0 [11] according to centers’ preference.

### 2.3. Atezolizumab and Bevacizumab

Atezolizumab and bevacizumab are approved by the European Medicines Agency (EMA) and the United States Food and Drug Administration (FDA) for the treatment of patients with HCC who have not yet received systemic treatment for HCC. The recommended doses are 1200 mg for atezolizumab and 15 mg/kg for Bevacizumab every three weeks. Treatment with the two drugs and discontinuation were performed according to the recommendations of the manufacturer and at the discretion of the treating physician.

### 2.4. Statistical Analysis

The present study is as a retrospective cohort study. All patients were followed until death or last contact. The primary end point was overall survival (OS), the secondary end points included progression-free survival (PFS), response rate, occurrence of bleeding complications, and safety.

Data on baseline characteristics, radiological response, and adverse events were summarized using descriptive statistics. Continuous variables are shown as median and full range, and categorical variables are reported as frequencies and percentages. Median duration of therapy was defined as the time from the first administration until the last administration of the drugs. Patients who still received atezolizumab with or without concomitant bevacizumab at data cut-off were censored. Patients with at least one staging imaging assessment were evaluated for radiological response.

Data from patients, who died without radiologically confirmed tumor progression, were censored at the date of the last radiological assessment or death. Progression-free survival (PFS) was defined as the time from the date of the first therapy administration until radiological disease progression or death, whatever occurred first. Patients still alive and without radiologically confirmed progression at the date of the last contact or data cut-off were censored. Overall survival (OS) was defined as the time from the start of the treatment with atezolizumab and bevacizumab until the date of death. Survival curves were calculated with the Kaplan–Meier method and compared by means of the log-rank test. To analyze prognostic parameters uni- and multivariable Cox regression models with forward stepwise likelihood ratio were used. Statistical analyses were performed with SPSS (Version 27.0, IBM, New York, NY, USA) and GraphPad Prism 8.0 (GraphPad Software, La Jolla, CA, USA). Differences between different patient cohorts were determined using the nonparametric Wilcoxon–Mann–Whitney and Kruskal–Wallis tests or Fisher’s exact text. For the sub-analysis of multiple comparisons, the Bonferroni correction was used. *p* values < 0.05 were considered significant.

## 3. Results

### 3.1. Patients

Sixty-six patients from four centers (one Austrian center, and three German centers) were included. Data cut-off for the analysis was 15 September 2021. Fifty-four patients (82%) were male, and the median age was 66 years (range 30–89 years). Additional baseline characteristics are shown in Table 1. Most patients had compensated cirrhosis (n = 34; 52%), while Child–Pugh class B cirrhosis was observed in 23 patients (35%), and class C cirrhosis in 5 patients (8%). For 4 patients, Child–Pugh assessment was not reported.

The median follow-up time was 211 days, with a range of 1 to 995 days from atezolizumab and Bevacizumab treatment initiation. Twenty-seven patients had died at the date of data cut-off.

### 3.2. Efficacy

At last contact, 51 patients (77%) had stopped atezolizumab and bevacizumab treatment. Thirteen patients (20%) were still on treatment, and two patients (3%) were lost to follow-up. The median time of treatment was 110 days (+995 days, 33 month), and the median number of cycles administered to the patients was 21. Overall, 52 patients (79%) had at least one follow-up imaging for the assessment of tumor response. Best responses included complete response (CR) in 7 patients (11%), partial response (PR) in 12 patients (18%), stable disease (SD) in 22 patients (33%), and progressive disease in 11 patients (17%) (Table 2); for 14 patients (21.1%), staging at data analysis was not available, and therefore, they were not evaluable for best response. The median progression-free (PFS) survival was 6.5 months (95% confidence interval (CI) of 4.0–9.1 months) (Figure 1A).

Patients with HCC due to viral hepatitis had a more favorable PFS (median PFS 17.3 months, 95% confidence interval (CI) of 5.6–29 months) than patients without a history of viral hepatitis (median PFS 6.1 months, 95% confidence interval (CI) of 3.1–8.9 months), corresponding to a hazard ratio (HR) of 0.48 with a 95% CI of 0.24–0.99 (*p* < 0.05) (Figure 1B).

Median overall survival (OS) was not reached in this cohort (6-month OS: 69%, 12-month OS: 60%, 18-month OS: 58%) (Figure 2A).

Patients with compensated liver disease (Child Pugh A) had a much more favorable prognosis than patients with more advanced liver disease (Child Pugh B and C cirrhosis). The 12-month survival rate in patients with Child A cirrhosis was 78%. (Figure 2B).

Patients with viral hepatitis tended to have a more favorable prognosis (median OS not reached) than patients without viral-related HCC (median OS 11.8 months, 95% confidence interval (CI) of 9.4–14.7 months), HR 0.61 with a 95% CI of 0.26–1.45 (*p* > 0.2) (Figure 2C).

### 3.3. Safety

At the time of data cut-off, 44 of 66 patients (67%) had stopped the treatment with atezolizumab and bevacizumab. In 25 patients (38%), bevacizumab was paused during treatment, and atezolizumab therapy was continued. Fifty-one (77%) of all patients reported at least one adverse event, and 39 patients (59%) experienced a high-grade (grade 3 or higher) event. The most common adverse events were bleeding events in 30.3% of the patients, worsening of renal function in 15.2%, and ascites in 8 patients (12.1%). Seven (10.6%) patients were diagnosed with variceal bleeding after treatment initiation. Of all patients with documented baseline variceal status (n = 55; 83%), 12 patients had grade 1, and 16 patients had varices above grade 1 (13 patients with grade 2, and 3 patients with grade 3 varices). None of the patients underwent prophylactic ligation therapy at baseline in this real-world cohort.

In total, 38 patients were administered a betablocker at the study start (59.1%), whereas 27 (40.9%) were not. Eleven of 38 (28.9%) patients received a selective betablocker, mostly prescribed for past cardiovascular reasons (bisoprolol and nebivolol), while 27 (71.1%) received a non-selective betablocker (19 were administered carvedilol, and 8 patients propranolol) explicitly for variceal bleeding prevention. Eighteen of the 27 patients without betablocker did not present diagnosed varices, while 9 patients with diagnosed varices used a betablocker. There was no statistical difference in betablocker use in patients with variceal bleeding compared to patients without bleeding; however, two patients with a bleeding event did not use betablockers.

Bleeding events were associated with the stage of varices (*p* < 0.05). In a multivariable analysis, the stage of varices was the only significant risk factor for bleeding, whereas betablocker intake, ALBI score, MELD score, and Child–Pugh score where not significantly associated with the risk of bleeding. For a detailed list of adverse events, see Table 3.

### 3.4. Factors Associated with Survival

Known predictors of survival in patients with HCC include liver function and AFP levels.

The variables gender, age (≤65 years vs. >65 years), hepatotropic virus infection, BCLC stage, AFP levels (≤400 ng/mL vs. >400 ng/mL), albumin–bilirubin (ALBI) score, Child–Pugh score, extrahepatic spread, and prior therapy were included in a multivariable model. Furthermore, all factors from univariable analysis with a *p*-value < 0.1 were included in the multivariable model. As shown in Table 4, Child–Pugh A cirrhosis and prior local therapy were independently associated with OS.

## 4. Discussion

Atezolizumab and bevacizumab have become the new standard of care for the first-line treatment of advanced HCC. We report our first experience with atezolizumab and bevacizumab in real-life European patients. Treatment was feasible and effective. The median PFS of 6.5 months was similar to that reported in the pivotal phase 3 trial (6.8 months) [4]. Furthermore, the 12-month survival rate in our patients with compensated liver disease was even better than in the patients in the trial (78% versus 67%). The subgroup of patients with viral hepatitis, namely, hepatitis B or hepatitis C, had a more favorable prognosis than patients without a history of viral hepatitis concerning PFS and OS. Also an exploratory subgroup analysis of the IMbrave150 trial favored immunotherapy for patients with viral hepatitis [12]. The high efficacy of atezolizumab and bevacizumab compared to sorafenib was also supported by the analysis of Chinese patients treated in the IMbrave150 trial [13]. In this group of patients, more than 90% had a history of viral hepatitis infection, mainly hepatitis B; 77% of the patients were alive 12 months after treatment initiation. Treatment response in our real-life cohort was 29%, which is nearly the same as in the phase 3 trial, which reported a treatment response of 27% [4]. In our multivariable analysis, only Child–Pugh stage A cirrhosis and prior local therapy were independently associated with survival. In support of these findings, emerging data are showing less effectiveness of immunotherapy in NASH/NAFLD patients with HCC, most probably due to an altered immune environment [14,15].

Comparable real-world data on atezolizumab/bevacizumab are scarce. Iwamoto and colleagues retrospectively analyzed 61 patients from Japan and found a median PFS of 5.4 month, a disease control rate of 86.3%, and adverse event rates of any grade and higher than grade 3 of 98% and 29.4%, respectively. However, 23 patients (62.7%) in this study underwent at least one line of prior small-molecule treatment, hampering a direct comparison [16].

Another group from Japan published real-world data concerning tumor response and safety for atezolizumab/bevacizumab in 40 patients with Child–Pugh A cirrhosis [17]. Twenty-four patients had a previous treatment experience with molecular agents (TKI). They found an ORR of 22.5% based on mRECIST. Multivariate analysis showed that an AFP ratio <1.0 at 3 weeks (odds ratio 39.2, 95% confidence interval CI 2.37–649.0, *p* = 0.0103) was the only significant factor for predicting an early response.

Hiraoka et al., aimed at the evaluation of early response (6 weeks) to atezolizumab/bevacizumab and included 171 HCC patients from Japan; again, 96 patients were systemically pretreated [18]. In initial imaging examination findings, they described objective response rates for early tumor shrinkage and disease control after 6 weeks (ORR-6W/DCR-6W) of 10.6% and 79.6%, respectively. Hayakawa et al., published a short report describing 52 patients undergoing atezolizumab/bevacizumab treatment (only 23 receiving it as first-line treatment). They found an objective response rate (ORR) and disease control rate (DCR) in all patients of 15.4% and 57.7%, respectively, and suggested AFP response as a predictive marker [19].

Sho et al., investigated 64 patients, 46 of whom 46 (71.9%) did not meet the inclusion criteria of the IMBRAVE 150 trial; 44 of these 46 patients where systemically pretreated. They showed good safety and efficacy for these patients. Interestingly, none of the 15 patients with hepatitis B experienced progressive disease [20].

Liver function is well known to be highly prognostic for survival in patients with HCC [21]. We recently reported that patients with more advanced cirrhosis receiving nivolumab obtain only a marginal benefit from a tumor-specific treatment and have a poor overall prognosis [22,23]. Therefore, immunotherapy seems to be of value only for patients with a relatively well-preserved liver function [24,25]. Interestingly, the 12-month survival in patients with Child B cirrhosis was still 39%, indicating that a subgroup of these patients does benefit from a tumor-specific treatment. These could be patients whose liver function impairment is mainly driven by a large intrahepatic tumor load.

In our cohort, patients with non-viral HCC tended to have a worse outcome compared to patients with viral-related HCC, which, however, was not significant in the uni- or multivariate analysis. Recently, Pfister et al., published highly discussed evidence of lowered effectiveness of immunotherapy in HCC patients than in non-alcoholic steatohepatitis (NASH) patients with HCC due to the presence of special resident-like activated CD8^+^ cells in patients with NASH [15]. The poorer response of NASH patients is supported by findings of Inada et al., and others [14,26]. A meta-analysis of the two first-line trials that used sorafenib as the comparator (CheckMate 459 trial and IMbrave150 trial) [3,4] showed the same trend (non-viral HCC OS HR = 0.94, but HBV OS HR = 0.65 and HCV OS HR = 0.60).

Patient-reported outcomes of the IMBRAVE 150 trial were published separately [6] and showed benefits in terms of patient-reported quality of life, functioning, and disease symptoms with atezolizumab plus bevacizumab compared with sorafenib. In the pivotal trial, 98% patients with any AE were reported, and 56.5% of them had higher-than-grade 2 events, which corresponds to our data. We documented treatment discontinuation (complete, or stopping, or discontinuation of bevacizumab) due to an AE in 20 patients (30%), which fits the trial data as well. In the trail, 7% of the patients reported bleeding, while our event rate was higher, most probably due to a less strict patient inclusion for treatment in real life and the overall worse liver function in our patients. As recently published, an important adverse event seems to be hypertension in atezolizumab/bevacizumab-treated patients (up to 30% of patients with grade 3) [27], which was reported at a much lower frequency in our cohort, and this may implicate underdiagnosis in real-life practice.

## 5. Conclusions

The combination of atezolizumab and bevacizumab is highly effective for patients with hepatocellular carcinoma in real life. Patients with cirrhosis and hepatocellular carcinoma of viral origin seemed to respond better than those with non-viral HCCs. Variceal bleeding is an important adverse event of this drug combination. In patients with compromised liver function (Child–Pugh B and C cirrhosis), the drug combination showed low efficacy; therefore, treatment for these patients should be well considered.

## Figures and Tables

**Figure 1 cancers-14-01722-f001:**
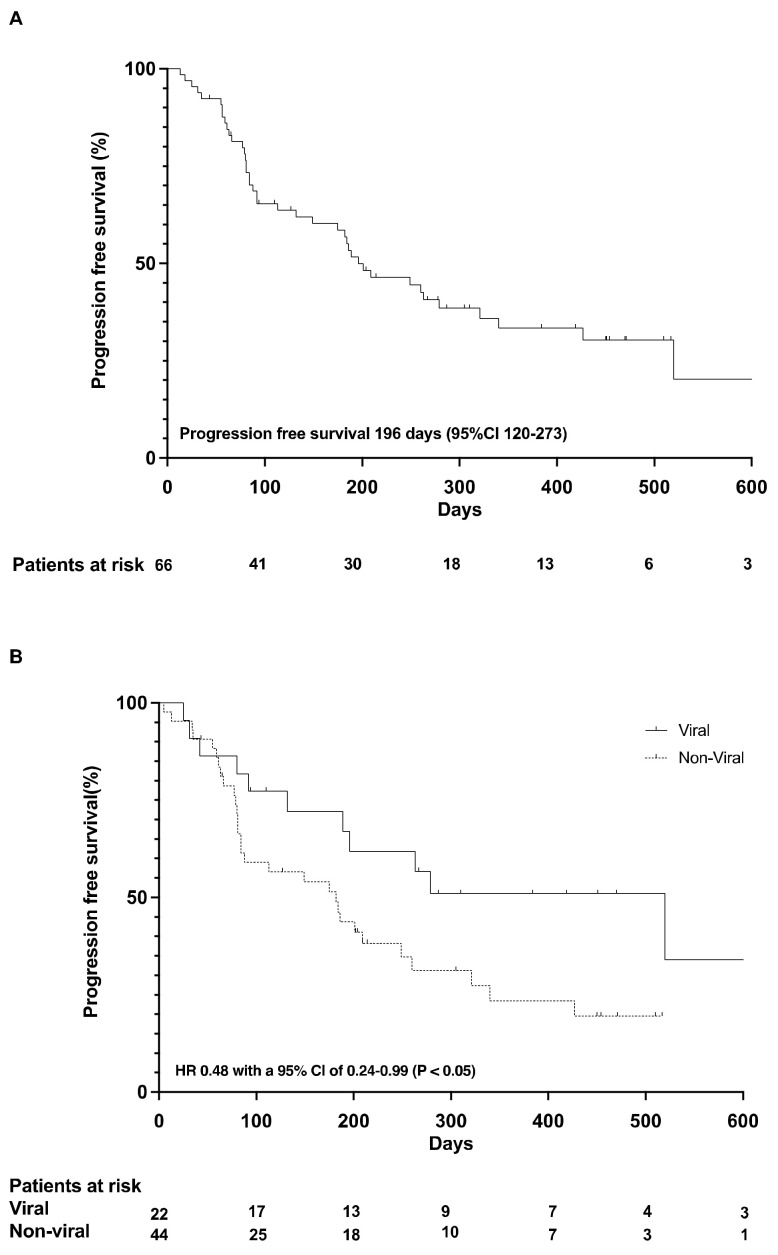
Progression free survival (**A**); PFS according to viral and non-viral etiology (**B**).

**Figure 2 cancers-14-01722-f002:**
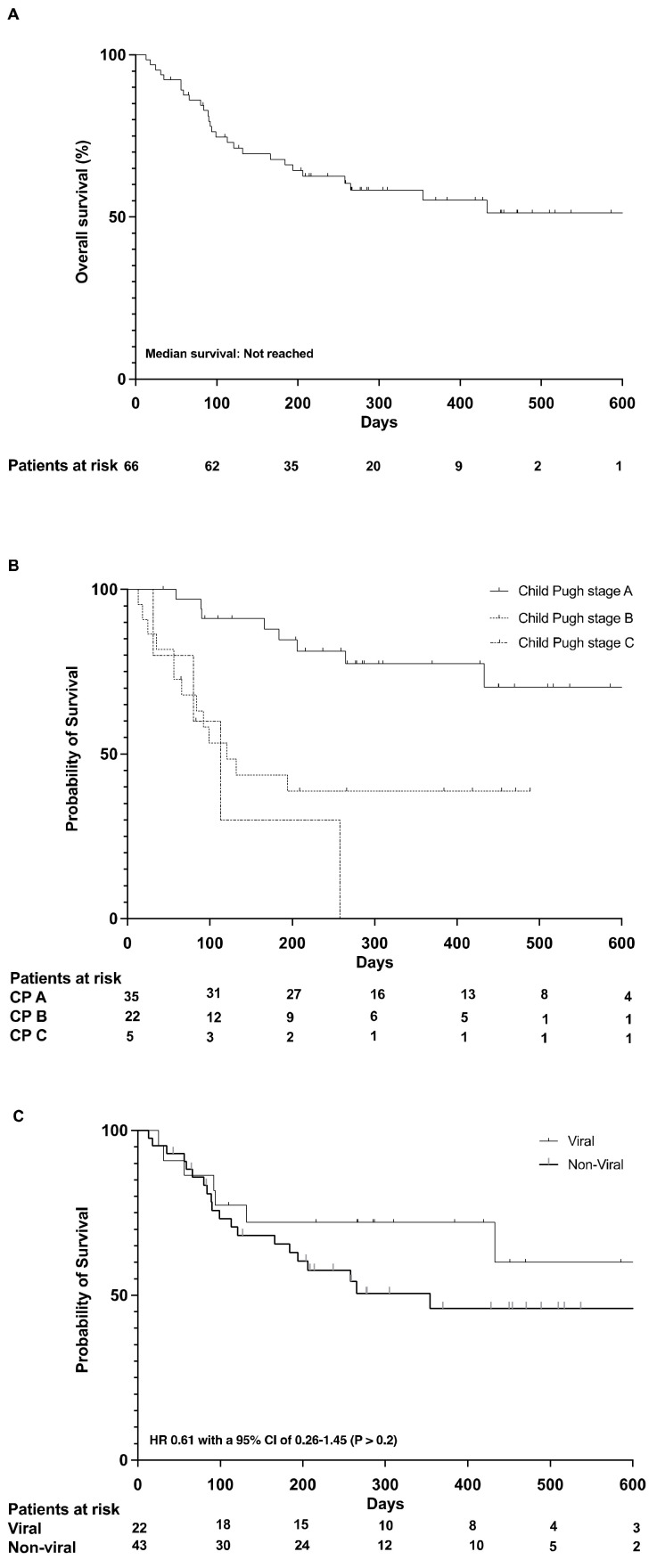
Overal survival (**A**); Survival according to Child Pugh stage (**B**); Survival according to viral and non-viral etiology (**C**).

**Table 1 cancers-14-01722-t001:** Patient characteristics.

Parameter	Patients
Epidemiology	
Patients, n	66
Gender, m/f (%)	54/12 (81.8/18.2)
Age, median, range	65 (30–88)
Etiology of liver disease	
Alcohol, n (%)	25 (37.9)
Hepatitis C, n (%)	14 (21.2)
Hepatitis B, n (%)	9 (13.6)
NASH/NAFLD ^1^, n (%)	18. (27.3)
BCLC stage ^2^	
A, n (%)	1 (1.5)
B, n (%)	22 (33.3)
C, n (%)	35 (53.0)
D, n (%)	8 (12.1)
MVI ^3^, n (%)	29 (43.9)
EHS ^4^, n (%)	18 (27.3)
Child–Pugh score	
A, n (%)	35 (53.0)
B, n (%)	23 (34.8)
C, n (%)	5 (7.6)
Albumin–Bilirubin (ALBI) grade	
1, n (%)	14 (21.2)
2, n (%)	46 (69.7)
3, n (%)	6 (9.1)
MELD ^5^, median, range	10 (6–23)
Betablocker medication, n(%)	38 (59.1)
Prior Treatment	
Resection, n (%)	9 (13.6)
Local ablation *, n (%)	11 (16.7)
Loco-regional (TACE/SIRT) ^6^, n (%)	27 (40.9)
Laboratory results	
BMI ^7^, median, range	27.6 (16.9–42.5)
ALT ^8^ (U/L), median, range	42 (7–1260)
AST ^9^ (U/L), median, range	64 (10–876)
Bilirubin (mg/dL), median, range	1.5 (0.2–9.4)
Albumin (g/dL), median, range	3.2 (1.8–4.4)
INR ^10^, mean, median, range	1.27 (0.69–2.99)
CRP ^11^ (mg/dL), median, range	1.1 (0.15–10.9)
AFP ^12^ (ng/mL), median, range	17.65 (1–49220)
AFP > 400 ng/mL, n (%)	19 (28.8)

Abbreviations: ^1^ NASH, non-alcoholic steatohepatitis; ^2^ BCLC, Barcelona liver clinic; ^3^ MVI, Macrovascular invasion; ^4^ EHS, Extrahepatic spread; ^5^ MELD, model of end-stage liver disease; ^6^ TACE/SIRT, transarterial chemoembolization/selective internal radiotherapy; ^7^ BMI, Body Mass Index; ^8^ ALT, alanine aminotransferase, ^9^ AST, aspartate aminotransferase; ^10^ INR, internationalized ratio; ^11^ CRP, C-reactive protein; ^12^ AFP, alpha-Fetoprotein. * including radiofrequency ablation (RFA), microwave ablation (MWA).

**Table 2 cancers-14-01722-t002:** Radiological response and survival data.

Parameter	Patients
Best documented response	
Complete response (CR), n (%)	7 (11.0)
Partial response (PR), n (%)	12 (18.0)
Stable disease (SD), n (%)	22 (33.0)
Progressive disease (PD), n (%)	11 (17.0)
Not evaluable (NA), n (%)	14 (21.0)
Disease control rate (DCR), (%)	62.0%
PFS ^1^, median (95%CI), month	6.5 (4.0–9.1)
OS ^2^, median days (95%CI), month	Not reached

Abbreviations: ^1^ PFS, progression-free survival; ^2^ OS, overall survival.

**Table 3 cancers-14-01722-t003:** Documented adverse events.

	Any Grade, (n/%)	≥Grade 3, (n/%)	Leading to Any Treatment Discontinuation (n/%)	Leading to Death, (n/%)
Bleeding events	20 (30.3)	18 (27.3)	11 (16.7)	3 (4.5)
Gastrointestinal bleeding Variceal bleeding	14 (21.2) 7 (10.6)	14 (21.2) 7 (10.6)	6 (9.1) 5 (7.5)	1 (1.5)
Subarachnoidal hemorrhage	2 (3.0)	2 (3.0)		2 (3.0)
Epistaxis	4 (6.0)	2 (3.0)		
Worsening of renal function	10 (15.2)	8 (12.1)		
Acute kidney failure	7 (10.6)	7 (10.6)	1 (1.5)	
Acute on chronic kidney failure	1 (1.5)	1 (1.5)	1 (1.5)	
Ascites	8 (12.1)	6 (9.1)	1 (1.5)	
Pruritus	6 (9.1)			
Diarrhea	5 (7.6)	2 (3.0)		
Rash	4 (6.1)			
Fatigue	4 (6.1)			
Hyponatremia	3 (4.5)	1 (1.5)		
Arterial hypertension	3 (4.5)			
Ulcus lower extremities	3 (4.5)	1 (1.5)		
Acute on chronic liver failure	2 (3.0)	2 (3.0)	1 (1.5)	
Hepatic encephalopathy	2 (3.0)	2 (3.0)		
Allergic reaction	2 (3.0)	1 (1.5)	1 (1.5)	
Nausea	2 (3.0)			
Emesis	1 (1.5)			
Cholangitis	1 (1.5)	1 (1.5)	1 (1.5)	
Pyrexia	1 (1.5)	1 (1.5)		
Transient ischemic attack	1 (1.5)	1 (1.5)	1 (1.5)	
Pulmonary embolism	1 (1.5)	1 (1.5)	1 (1.5)	
Flare of autoimmune disease	1 (1.5)			
Insomnia	1 (1.5)			
Hyperbilirubinemia	1 (1.5)		1 (1.5)	
Spontaneous bacterial peritonitis	1 (1.5)	1 (1.5)		
Cough	1 (1.5)			
Hyperkalemia	1 (1.5)			
Hoarseness	1 (1.5)			
Vasculitis	1 (1.5)			
Anemia	1 (1.5)	1 (1.5)		
Proteinuria	1 (1.5)			
Edema	1 (1.5)			
Worsening of heart failure	1 (1.5)			
Stomatitis	1 (1.5)			
Nephritis	1 (1.5)			
Dry skin	1 (1.5)			
Immune checkpoint-inhibitor hepatitis (ICI)	1 (1.5)	1 (1.5)	1 (1.5)	
Esophageal candidiasis	1 (1.5)	1 (1.5)	1 (1.5)	
Urogenital abscess	1 (1.5)	1 (1.5)	1 (1.5)	

**Table 4 cancers-14-01722-t004:** Univariate and multivariate analyses of parameters associated with overall survival.

	Univariate Analysis			Multivariate Analysis		
Parameter	HR	95% CI	*p* Value	HR	95 % CI	*p* Value
Male gender	1.609	0.555–4.662	0.381			
Age < 65 years	0.645	0.302–1.380	0.259			
Viral Hepatitis	0.612	0.258–1.449	0.264			
BCLC AB	0.632	0.276–1.445	0.277			
AFP < 400 ng/mL	0.928	0.402–2.142	0.861			
ALBI score 1	0.043	0.005–0.337	0.003			
Child Pugh A	0.152	0.045–0.515	0.002	0.112	0.024–0.534	0.006
Extrahepatic spread of HCC	2.182	0.996–4.780	0.051			
Prior local therapy/surgery	0.450	0.205–0.988	0.047	0.346	0.122–0.978	0.045

Abbreviations: HR, hazard ratio; CI, confidence interval; BCLC stage AB. Barcelona Clinic Liver Cancer stage A and B; AFP, alpha-fetoprotein, ALBI score, albumin–bilirubin score.

## Data Availability

Data sharing not applicable. No new data were created or analyzed in this study.

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
