# Peer review of "Efficacy and Safety of Atezolizumab and Bevacizumab in the Real-World Treatment of Advanced Hepatocellular Carcinoma: Experience from Four Tertiary Centers"

_cancers, 2022, doi:10.3390/cancers14071722_

Round 1

Reviewer 1 Report

Line 227-8, please adjust according to new analysis (response to treatment not included)

Line 296-7, this important statement to should be included in the abstract

Author Response

We thank reviewer 1 again for the overall positive vote on this manuscript.

Line 227-8, please adjust according to new analysis (response to treatment not included)

à We adjusted the text in discussion text according to the survival data from revision round 1.

Line 296-7, this important statement to should be included in the abstract

à We added this to the abstract.

Reviewer 2 Report

The Authors have put a great effort into ameliorating their manuscript. Most of my suggestions were also addressed and I thank the Authors for considering my points.
I still have a concern about variceal bleeding prophylaxis, as this is a very important and debated topic in the real-life management of bevacizumab. The Authors provided an answer about mechanic ligation but dodged questions about nonselective beta-blockers. I still think that this point is key in the management of the therapy and expected by the readers of a high-tier journal. Also, a number of minor points deserve attention: 

1) Abstract: "The real-world PDF": Maybe the Authors intended PFS. 

2) Results - Efficacy: "Median time on treatment was 110 days (+ 995 days, 33 month. The.." Please check this sentence

3) Results: Data presented in lines 166-180 do not match with the results presented in Table 4. I figure that the authors are reporting univariable results in the text, but this fact should be clearly illustrated, otherwise readers might be confused.

4) Discussion, line 264-267: I agree that this point deserves discussion and even a trend should be reported in a study with limited sample size. But, again, this trend was not confirmed in multivariable analysis and this fact should also be reported in this sentence.

Author Response

We thank the reviewer for this positive comment. As we agree with the reviewer that the data concerning the betablockers is important we reviewed all patient charts for betablocker use at the moment A+B was started.

38 patients had a betablocker (59.1%) and 27 (40.9%) had none. 11 of 38 (28.9%) had a selective betablocker, mostly for cardiovascular reasons (bisoprolol and nebivolol) while 27 (71.1%) used an non-selective betablocker (19 had carvedilol and 8 patients’ propranolol).

18 of the 27 patients without betablocker did not have diagnosed varices, while 9 patients with diagnosed varices used a betablocker

There was no statistical difference in betablocker use in patients with variceal bleeding compared to patients without bleeding, however 2 patients with a bleeding event did not use betablockers.

We added this information to the manuscript. Betablocker use or non-use was not significantly associated with bleeding events.

  • Abstract: "The real-world PDF": Maybe the Authors intended PFS. 

à We corrected the typo.

  • Results - Efficacy: "Median time on treatment was 110 days (+ 995 days, 33 month. The.." Please check this sentence

 à We corrected the typo.

  • Results: Data presented in lines 166-180 do not match with the results presented in Table 4. I figure that the authors are reporting univariable results in the text, but this fact should be clearly illustrated, otherwise readers might be confused.

àWe agree with you, this sounds confusing as it was written. We deleted the last part here, as results from uni-and multivariate analysis are reported later on.

Discussion, line 264-267: I agree that this point deserves discussion and even a trend should be reported in a study with limited sample size. But, again, this trend was not confirmed in multivariable analysis and this fact should also be reported in this sentence.

à We agree, this is highly discussable and we made it clear, that this is only speculation from the univariate analysis

Reviewer 3 Report

This study has major flaws due to the fact that >40% of patients were Child B or C, and this modifies the OS results for the whole cohort, thereby limiting data interpretation.

When analyzing prognostic factors for OS, the authors reports in Table 4 results that are completely different from those in the text (lines 206-207).

The discussion should be trimmed.

Author Response

This study has major flaws due to the fact that >40% of patients were Child B or C, and this modifies the OS results for the whole cohort, thereby limiting data interpretation.

When analyzing prognostic factors for OS, the authors reports in Table 4 results that are completely different from those in the text (lines 206-207).

The discussion should be trimmed.

à Thank you again for reviewing the manuscript. We corrected the part in “Factors associated with survival” according to the new multivariate analysis.

In discussion we deleted the last part about cost effectiveness for shortening the discssuion.

Round 2

Reviewer 2 Report

The Authors addressed all of the raised points adequately. I have no further comments.

This manuscript is a resubmission of an earlier submission. The following is a list of the peer review reports and author responses from that submission.

Round 1

Reviewer 1 Report

The article by Himmelsbach and colleagues deals with a very important topic related to the approval in the real life of atezolizumab and bevacizumab (A+B) for advanced/unresectable HCC. However, I am deeply concerned by several methodological flaws inherent to this research. In turn, these may generate wrong conclusions misleading the reader on the efficacy of A+B.

Major comments:

1) Lines 60-62: A+B were approved in EU only in November 2020. Thus, the Authors should explain the context of their retrospective cohort study: was it a clinical trial? An expanded access program? If so, were inclusion/exclusion criteria considered?

2) Lines 119-120: How was it possible to treat Child and Child C patients? This would be a major protocol violation in any clinical study, and dramatically flaws the results of the entire study as almost half of the patients (43%) would be excluded from the ImBrave 150 study.

3) Figure 2b depicts a comparison which is not appropriate.

4) Table 4 is assumed to report on prognostic factors for OS, however prior treatments (local therapies) and extrahepatic spread were not included despite their prognostic significance: why?

Author Response

The article by Himmelsbach and colleagues deals with a very important topic related to the approval in the real life of atezolizumab and bevacizumab (A+B) for advanced/unresectable HCC. However, I am deeply concerned by several methodological flaws inherent to this research. In turn, these may generate wrong conclusions misleading the reader on the efficacy of A+B.

Dear Reviewer thank you for your comments to our work, we changed the manuscript accordingly (see our answers below). As this is a retrospective analysis we agree that there are some inherent flaws, however in our opinion the strength is based on the multicenter aspect and that these are patients as they are seen in daily practice and therefore represents the “real-life”.

Major comments:

  • Lines 60-62: A+B were approved in EU only in November 2020. Thus, the Authors should explain the context of their retrospective cohort study: was it a clinical trial? An expanded access programs? If so, were inclusion/exclusion criteria considered?

Our study consists of all patients retrospectively included in the outpatient’s clinics of the named centers. Several patients were treated before the EMA approval, this is due to exceptional approvals by the health care insurance companies to cover the costs. It is common practice to apply for this, when there is promising data for treatments in patients which have no better options. In case of A+B this was very much the case as there was very early data of good efficiency. As the patients were treated in daily practice there were no distinct inclusion/exclusion criteria, however, this make these kinds of studies somehow important, as it reflects the patients who are treated in real life. We included this information in M&M. These were not trial patients!

  • Lines 119-120: How was it possible to treat Child and Child C patients? This would be a major protocol violation in any clinical study, and dramatically flaws the results of the entire study as almost half of the patients (43%) would be excluded from the ImBrave 150 study.

This connects to the comment above. These were real-life patients and we know from several other published works (e.g. Scheiner et al., J Hep 2021, > 30% Child B and  up to 10% Child C), that it is common practice to treat HCC patients with advanced liver disease despite the trial data, which mostly include only Child A patients. The new German S3 HCC guideline suggests to try a treatment in patients up to Child B8.

From recent real-world data concerning other HCC treatments however Child B patients seem to perform much worse compared to Child A patients, therefore this is important data in our opinion.

  • Figure 2b depicts a comparison which is not appropriate.

You are right, this was an editing error, we deleted this.

4) Table 4 is assumed to report on prognostic factors for OS, however prior treatments (local therapies) and extrahepatic spread were not included despite their prognostic significance: why?

Mostly the extrahepatic spread is somehow included in the BCLC staging, but this is a very good suggestion, so we did a re-calculation in uni-variate analysis and multivariate analysis including these two and found a significant result for both (lower risk for death for pretreated patients and higher risk for patients with EHS, p only 0.051. The results of the multivariate changed and as independent factors the pretreatment and Child A stadium and pretreatment remained.

We included this in the table 4.

Reviewer 2 Report

I read with pleasure this paper about a real-life experience with atezolizumab-bevacizumab (A+B) for hepatocellular carcinoma (HCC). The topic is of interest, but there are critical points which should be addressed, as the methodology which is currently used produced biased results.

CRITICAL POINTS
 1) Patients who died before performing the first radiological assessment are usually counted as progressors, especially if death occurred in the setting of clinical progression. Cosnequently, the Authors found an astonishingly low rate of progressive disease (17%) compared to other studies.
This choice also has some odd consequences, for instance:
 - Abstract: "Best responses included complete response (CR) in 7 patients (11%), partial response (PR) in 12 patients (18%), stable disease (SD) in 22 patients (33%), and progressive disease in 11 patients (17%), respectively." The total is not 100%
 - PAge 4, line 136. Best  responses included complete response (CR) in 7 patients (11%), partial response (PR) in 12 patients (18%), stable disease (SD) in 22 patients (33%), and progressive disease in 11 patients (17%), respectively".Same problem.

Also, the Authors stated that:
 - Page 3, line 98: "Data from patients, who died without radiologically confirmed tumor progression, were censored at the date of last radiological assessment or death". This choice is not correct to define PFS, rather it can be used to calculate time-to-progression (TTP). Death befoer imaging counts as an event in the PFS analysy. Please report the correct values of PFS after correcting for this factor.

 2) Radiological response was inserted in the multivariable Cox regression as a standard variable. This choice generate an immortal-time bias, as only patients who lived enough time to reach this milestone could be inserted in the analysis. Instead, early progressors have been discarded. This way, the analysis produce an inflated effect of radiological response. If the Authors want to keep radiological response in the Cox regression, they should consider it as a time-dependent variable, and handle it accordingly. This paper is an example of how to handle time-dependent variables in HCC systemic treatments: PMID: 24703956

MAJOR POINTS
 1) It would be interesting to know how many patients with advanced HCC were not eligible to receive A+B (and thus received TKI monotherapies) in the Centers in the timeframe of the study. The actual eligibility to combination therapies is a topic of great interest and a real-life study can provide insightful information.

 2) Pag 2, line 75: "Side effects were recorded at every visit and graded according to the Common Terminology Criteria for Adverse Events (CTCAE) version 4.0 or 5.0 according to centers preference." Did the Authors verify whether some key adverse events (for instance, thromboembolic events and bleeding events) did not change their grading from version 4.0 to 5.0?

 3) Page 9, line 175. "Of all patients with documented baseline variceal status (n=55; 83%), 12 patients had grade 1 and 16 patients had varices above grade 1 (13 patients with grade 2 and 3 patients with grade 3). Bleeding events were associated with the stage of varices (P < 0.05)". Please report also how many patients with varices received a prophylaxis either with mechanic ligation or non-selective beta-blockers before initiating A+B. Also, verify whether prophylaxis reduced the risk of bleeding in your cohort.

MINOR POINTS 

1) paG 2, LINE 43: different font

2) paG 2, LINE 49: "und" instead of "and

3) pag 2, LINE 53. Missing reference. The Authors can use PMID: 31671581 

4) Page 3, line 115: The sentence start with a number. Please use "sixty-six" in letter. Same problem in line 116, 122, 133.

5) Table 1: albumin is reported as mg/dl, instead of g/dl.

6) Page 7, line 161: "am more favorable prognosis". Please correct.

7) Table 3: please define ICI-hepatitis

Author Response

Reviewer 2:

I read with pleasure this paper about a real-life experience with atezolizumab-bevacizumab (A+B) for hepatocellular carcinoma (HCC). The topic is of interest, but there are critical points which should be addressed, as the methodology which is currently used produced biased results.

We thank you for the positive comment on our work, please find our answers to your raised questions attached.

CRITICAL POINTS
 1) Patients who died before performing the first radiological assessment are usually counted as progressors, especially if death occurred in the setting of clinical progression. Consequently, the Authors found an astonishingly low rate of progressive disease (17%) compared to other studies.

Please see comments below for calculation of PFS, we did that accordingly. The low rate of progressive disease somehow connects to the follow-up time here. In the IMBRAVE trial it was nearly the same percentage with progressive disease (19.6%) compared to our 17% (only 66 patients and even shorter FU with 7 month in our trial)

This choice also has some odd consequences, for instance:
 - Abstract: "Best responses included complete response (CR) in 7 patients (11%), partial response (PR) in 12 patients (18%), stable disease (SD) in 22 patients (33%), and progressive disease in 11 patients (17%), respectively." The total is not 100%
 - PAge 4, line 136. Best responses included complete response (CR) in 7 patients (11%), partial response (PR) in 12 patients (18%), stable disease (SD) in 22 patients (33%), and progressive disease in 11 patients (17%), respectively". Same problem.

We agree with the reviewer that this is relevant problem. Our approach was to analyze the best documented response; however, we agree, that the data presentation may be misleading. As you calculated correctly the patients do not add up to 100% as we have probably written this in a misleading manner. These numbers are only the patients, where at least one staging/FU of 3 month after starting the therapy, was available. Therefore, we named this “best reported response”.

There were 14 patients (21.1%) without any staging dying or dropping out before this, where the “best documented response” is not available (NA). We did not specifically say this in the manuscript, which we corrected in the manuscript.

As it was the main goal of this work to find the effect of A+B in these patients, therefore we think the style of showing the data is appropriate. To our knowledge it is still a matter of discussion how to present this kind of data of retrospective nature dealing with dropouts before first staging.

However, patients who died are still counted as progressions in OS and PFS calculations as you mentioned correctly.

Also, the Authors stated that:
 - Page 3, line 98: "Data from patients, who died without radiologically confirmed tumor progression, were censored at the date of last radiological assessment or death". This choice is not correct to define PFS, rather it can be used to calculate time-to-progression (TTP). Death before imaging counts as an event in the PFS analysy. Please report the correct values of PFS after correcting for this factor.

The reviewer is right, this is written misleadingly. We deleted the sentence as it may mislead the reader, this was only about the censoring when death occurred (as we stated above). PFS is calculated as: Progression free survival (PFS) was defined as the time from the date of first therapy administration until radiological disease progression or death, whatever occurred first.

We already calculated PFS as you stated, therefore the numbers are right in our opionon, we discussed this again with our biostatistics institute to not make a mistake here. Furthermore PFS and OS results perfectly fit what to expect.

 2) Radiological response was inserted in the multivariable Cox regression as a standard variable. This choice generates an immortal-time bias, as only patients who lived enough time to reach this milestone could be inserted in the analysis. Instead, early progressors have been discarded. This way, the analysis produce an inflated effect of radiological response. If the Authors want to keep radiological response in the Cox regression, they should consider it as a time-dependent variable, and handle it accordingly. This paper is an example of how to handle time-dependent variables in HCC systemic treatments: PMID: 24703956

We absolutely agree with the reviewer here, this would discard 21% of the patients without staging. This is a comment of absolute importance, we thank you for this hint. We took this variable out of the analysis, the table 4 was corrected and extended with the comments from reviewer 1 (please see there as we re-calucated the whole analysis).

MAJOR POINTS
 1) It would be interesting to know how many patients with advanced HCC were not eligible to receive A+B (and thus received TKI monotherapies) in the Centers in the timeframe of the study. The actual eligibility to combination therapies is a topic of great interest and a real-life study can provide insightful information.

Unfortunately, we do not have this data at the moment and not for this manuscript. This would presume, that we included all patients treated with HCC at the different centers in this analysis, however only patients with A+B were included. There are not much contraindications for A+B (e.g. LTX after liver transplant), however, a manuscript dealing with this topic (systemic therapy in HCC post liver transplant) from our group is in preparation where we deal with this point.

 2) Pag 2, line 75: "Side effects were recorded at every visit and graded according to the Common Terminology Criteria for Adverse Events (CTCAE) version 4.0 or 5.0 according to centers preference." Did the Authors verify whether some key adverse events (for instance, thromboembolic events and bleeding events) did not change their grading from version 4.0 to 5.0?

Yes, this was investigated as much as it was possible in a retrospective analysis. As you can see in table 3 we found 3 patients with a bleeding event grade 5 (death) most probably attributable to the treatment.

 3) Page 9, line 175. "Of all patients with documented baseline variceal status (n=55; 83%), 12 patients had grade 1 and 16 patients had varices above grade 1 (13 patients with grade 2 and 3 patients with grade 3). Bleeding events were associated with the stage of varices (P < 0.05)". Please report also how many patients with varices received a prophylaxis either with mechanic ligation or non-selective beta-blockers before initiating A+B. Also, verify whether prophylaxis reduced the risk of bleeding in your cohort.

None of the included patients underwent planned prophylactic ligation before starting the therapy as far as we could evaluate the retrospective data. At least in the centers taking part in this study this was not standard of care at this time, in the meantime this is different all patients undergo baseline EGD and ligation therapy if varices above grade 2. We cannot compare effects of a prophylaxis due to this and because we do not have a comparison group.

Varices grade 1 are not eligible to prophylactic ligation therapy, however it is to be supposed, that variceal status can worsen under therapy with A+B. Patients with varices grade 3 should not be treated with A+B until litigation therapy is done, however, these patients still get treatment in clinical practice. This is important information we can derive from studies like this one in our opinion.

MINOR POINTS 

  • paG 2, LINE 43: different font

Most probably due to editorial changes, the font will be re-done from the editorial office.

  • paG 2, LINE 49: "und" instead of "and

Changed.

  • pag 2, LINE 53. Missing reference. The Authors can use PMID: 31671581 

We included this, thank you.

  • Page 3, line 115: The sentence start with a number. Please use "sixty-six" in letter. Same problem in line 116, 122, 133.

We changed this were we found numbers at the beginning of a sentence.

  • Table 1: albumin is reported as mg/dl, instead of g/dl.

Changed, your are right, the numbers a g/dl.

  • Page 7, line 161: "am more favorable prognosis". Please correct.

Changed.

7) Table 3: please define ICI-hepatitis

It means hepatitis due to immune checkpoint inhibitors (ICI), we have written this.

Reviewer 3 Report

Treatment with atezolizumab and bevacizumab has been shown to be superior to sorafenib in a randomized trial of highly selected patients, but the reproducibility of results in "real world" patients and the toxicity has been a matter or concern. This study show important data on efficacy and feasibility in non-selected patients. Similar studies have been published, but are in Asian populations and not directly comparable to the IMBrave study with respect to e.g. prior systemic treatment.

Due to its retrospective nature, details of toxicity and response rates have strong limitations, but the authors did mention these. Also the study only included  66 patients.

I have only af few suggestions and comments:

I would like the authors to include the number of patients treated with other kinds of systemic therapy for HCC in the same period of time at the insitutions, to illustrate the selection of patients.

I presume that the vital status of all patients was available (line 87). If not, it should be clearly stated.

In Tables, the number of patients with missing values should be outlined for each variable.

Line 134: Upper range of treatment duration is +995 days, not 995 (+33 months), not 33 months, as some were still on treatment.

Line 161: had a (not am)

Table 4. Please do not include treatment response in a multivariate prognostic model, as treatment response is not accessable before treatment initiation. Patients who respond should live longer! How many patients were included in the mulitivariate model (some variables were missing for some patients)?

Please provide an analysis of the prognostic value of a significant drop in AFP (for those with elevated values at baseline).

Line 249: Cohort, not cohor

Line 280: hepatocellular carcinoma, not hepatocellular

In the conclusion, it should be mentioned that patients with compromized liver function (CP C) did not seem to benefit.

Author Response

Treatment with atezolizumab and bevacizumab has been shown to be superior to sorafenib in a randomized trial of highly selected patients, but the reproducibility of results in "real world" patients and the toxicity has been a matter of concern. This study shows important data on efficacy and feasibility in non-selected patients. Similar studies have been published, but are in Asian populations and not directly comparable to the IMBrave study with respect to e.g. prior systemic treatment.

Due to its retrospective nature, details of toxicity and response rates have strong limitations, but the authors did mention these. Also, the study only included 66 patients.

Thank you for your positive comment on this small trial.

I have only a f few suggestions and comments:

I would like the authors to include the number of patients treated with other kinds of systemic therapy for HCC in the same period of time at the institutions, to illustrate the selection of patients.

As mentioned above, this data is not available. It is quite rare that patients in first line are treated with other therapies than A+B since the EMA approval, at least in our centers. The only “real” contraindication is HCC post-LTX (or severe autoimmune disease). Patients with post-LTX HCC are prepared in another manuscript from our multicenter cohort. As we saw from this study, even patients with the contraindication Child Pugh B (or even C) are treated at the consultant’s discretion. Therefore, as all patients with A+B treatment in the given time were included there is not a selection bias investigating this treatment. There were no A+B treated patients excluded or not investigated. (Please see comment above dealing with the same concern)

I presume that the vital status of all patients was available (line 87). If not, it should be clearly stated.

We could not exactly verify, what you mean here, the first sentence in statistical analysis? This was already commented above. The data on vital status is available, we tried to make this more clear.

In Tables, the number of patients with missing values should be outlined for each variable.

Missing values (e.g. in table 1) were quite rare in our cohort (if it happened, it was mostly 1-2 patients per variable), as adding this data would make the tables quite confusing and this is not relevant information for the calculations we left that out after discussing it. All variables, were missing data could be important (e.g. no staging) is mentioned in the text. We hope the reviewer can agree to this.

Line 134: Upper range of treatment duration is +995 days, not 995 (+33 months), not 33 months, as some were still on treatment.

We changed that, thank you

Line 161: had a (not am)

We changed that accordingly.

Table 4. Please do not include treatment response in a multivariate prognostic model, as treatment response is not accessible before treatment initiation. Patients who respond should live longer! How many patients were included in the multivariate model (some variables were missing for some patients)?

We re-calculated the whole uni- and multivariate analysis as this was right criticism from all reviewers, see comments above. None of these variables was missing, see comment above (only some lab values were missing in the whole data collection!)

Please provide an analysis of the prognostic value of a significant drop in AFP (for those with elevated values at baseline).

We do not have this data, as we did not collect information during the treatment course except treatment cycles, especially we do not have AFP levels during treatment. However, it is already known, that AFP response correlates with treatment response in HCC including a published metanalysis. To our experience AFP drop correlates quite good with treatment response.

Line 249: Cohort, not cohor

We changed that.

Line 280: hepatocellular carcinoma, not hepatocellular

We changed that.

In the conclusion, it should be mentioned that patients with compromized liver function (CP C) did not seem to benefit.

This is an important hint, we changed that accordingly.